# Physical Properties and Prebiotic Activities (*Lactobacillus* spp.) of Gelatine-Based Gels Formulated with Agave Fructans and Agave Syrups as Sucrose and Glucose Substitutes

**DOI:** 10.3390/molecules27154902

**Published:** 2022-07-31

**Authors:** Rogelio Rodríguez-Rodríguez, Paloma Barajas-Álvarez, Norma Morales-Hernández, Rosa María Camacho-Ruíz, Hugo Espinosa-Andrews

**Affiliations:** 1Department of Natural and Exact Sciences, Los Valles University Center (CUVALLES), University of Guadalajara, Highway Guadalajara-Ameca Km. 45.5, Ameca 46600, Mexico; rogelio.rodriguez4085@academicos.udg.mx; 2Food Technology Unit, Centre for Research and Assistance in Technology and Design of the State of Jalisco, AC (CIATEJ), Row Arenero 1227, El Bajío del Arenal, Zapopan 45019, Mexico; palomabal.14@gmail.com (P.B.-Á.); nmorales@ciatej.mx (N.M.-H.); 3Industrial Biotechnology, Centre for Research and Assistance in Technology and Design of the State of Jalisco, AC (CIATEJ) Row Arenero 1227, El Bajío del Arenal, Zapopan 45019, Mexico; rcamacho@ciatej.mx

**Keywords:** agave fructans, agave syrups, textural profile, prebiotic activity, *Lactobacillus* spp., functional foods

## Abstract

This research developed model foods of gelatine-based gels, where carbohydrates from *Agave tequilana* Weber var. Azul (agave syrups or/and agave fructans) were incorporated into gel formulations as healthy sucrose and glucose substitutes. The sugars (sucrose and glucose) were substituted by agave carbohydrates (agave syrups and agave fructans), obtaining the subsequent gel formulations: 100% agave syrup (F2 gel), 100% agave fructan (F3 gel), and 50% agave syrup–50% agave fructan (F4 gel). The unsubstituted gel formulation was used as a control (F1 gel). The prebiotic activities, physical properties, thermal stability (HP-TLC), and texture of gelatine-based gels were evaluated. The gel formulations showed translucent appearances with approximately 36 g/100 g of water and water activities values between 0.823 and 0.929. The HP-TLC analysis validated that agave fructans did not hydrolyse during the thermal process of gels production. Gels produced with agave syrup and agave fructan (F2-F4 gels) provided higher hardness, gumminess, and springiness values (*p* < 0.05) than those produced with glucose and sucrose (F1 gel). Gelatine-based gel formulations displayed prebiotic activities correlated to the ability of *Lactobacillus plantarum*, *Lactobacillus paracasei*, and *Lactobacillus rhamnosus* to use agave carbohydrates as carbon sources. Based on the prebiotic effect and physical and textural properties, the F2 and F4 gel formulations displayed the best techno-functional properties to produce gel soft candies.

## 1. Introduction

Currently, obesity, overweight, and diabetes are serious public health threats because they cause thousands of deaths and billion-dollar spending [1]. The World Health Organization (WHO) refers that more than 1.9 billion adults and 340 million children and adolescents aged between 5 and 19 years were overweight or obese in 2016. The principal causes were an excessive intake of energy-dense foods (high in fats and sugars) and decreased physical activity due to a sedentary lifestyle [2]. For many years, food scientists have been searching for ways to reduce fat, sugar, and salt in foods without losing their physicochemical, textural, and sensory properties, ensuring healthy and nutritious products for all consumers. For example, Belščak-Cvitanović et al. [3] developed chocolates with 20% fewer calories reducing the sucrose content with sweeteners such as inulin, oligofructose, and agave syrup, increasing the prebiotic activity and enhancing the organoleptic properties. Santiago-García et al. [4] reported that adding agave fructans to fat-reduced yogurts decreased the syneresis and increased water retention capacity and viscosity compared with the yogurt fat-reduced control. González-Herrera et al. [5] produced a snack from dehydrated apple supplemented with agave fructans exerting a probiotic effect by increasing short-chain fatty acid production in mice.

In Mexico, Agave genus plants are mainly used to produce alcoholic beverages such as tequila, mezcal, sotol, and bacanora. In recent years, the extraction of fructopolysaccharides from Agave species, known as agave fructans, has drawn attention in the food industry because agave fructans promote human health [6]. Fructans are prebiotic ingredients used technologically to replace the fat and sugar in foods. Moreover, they are consumed by gut microbiota, promoting human health through the production of short-chain fatty acids such as lactic acid, butyric acid, and propionic acid [7]. Fructans are oligosaccharides and polysaccharides of fructose units, used as reserve carbohydrates by plants such as chicory (*Cichorium intybus*), Jerusalem artichoke (*Helianthus tuberosus*), garlic (*Allium sativum*), onion (*Allium cepa*), asparagus (*Asparagus officinalis* L.) and *Agave* spp. Mexico has a wide variety of agave species; the most common sources of Agave are *A. tequilana* Weber var. azul, *A. angustifolia* Haw, and *A. salmiana* [8]. Mainly, *A. tequilana* Weber contains highly branched fructans with β-(2-1) and β-(2-6) linkages of several polymerisation degrees (DP, from 3 to 60 units of fructose) [9]. Agave fructans offer several technological properties for food technologists, including low-calorie sweeteners, fat replacers, soluble fibre, microencapsulation carriers, texture, and viscosity modifiers [6]. Furthermore, agave fructans give important human health support, e.g., agave fructans regulate the probiotic activity, glucose metabolism, obesity, and overweight [6,10].

Human gut microbiota comprises diverse microorganisms populations; mainly, the small intestine is colonised primarily by *Lactobacillaceae* and *Enterobacteriaceae* [11]. These population strains in the human gut depend on dietary habits, race, stress, medication, host genetics, toxin exposure, and pathogen invasion. It was reported that the *Lactobacillus plantarum*, *Lactobacillus rhamnosus*, and *Lactobacillus paracasei* are used to prevent and treat diverse intestinal diseases in human health [12,13]. In the host gut, the probiotic growth depends on the gut environment (pH, oxygen, and transit time), accessibility, and metabolism of the strains for a specific carbon source (known as the prebiotic effect). Michel-Barba et al. [14] reported that the carbon source used in developing functional food was essential to improving probiotic activity. The authors reported that the *L. plantarum* strain showed the fastest growth rate than *L. paracasei* or *L. rhamnosus* when solid corn syrups were used as a carbon source.

On the other hand, the enzymatic and thermal hydrolyses of the agave fructans are used to produce a natural sweetener of a low glycaemic index, known as agave syrups or agave nectars. These syrups are composed mainly of fructose and small amounts of glucose, sucrose, kestose, and nystose, the last two display prebiotic effects [15]. In this sense, agave carbohydrates (agave syrups and agave fructans) provide an excellent opportunity to develop new food products for health problems related to overweight, obesity, and diabetes.

Children prefer soft candies because of their texture, appearance, taste, and flavour [16]. Currently, consumers demand new products with healthier and low-calorie ingredients without negatively impacting product quality [17]. Different soluble fibres and natural sweeteners are used to replace or reduce sucrose and glucose contents in candies, including chicory inulin [16,18] and agave syrups [19]. Glucose and some polyols are commonly used to avoid the crystallisation of sucrose during the thermal process of candy production, controlling the water migration and favouring the texture properties in the final product. Delgado and Bañón [18] reported that the sugar substitution with inulin in gummy jelly products enhanced flavour and provided a slightly softer and springier texture. Čižauskaitė et al. [19] observed that replacing sugar with xylitol or agave syrups in gummies did not significantly change its textural properties (firmness, strengthless, and hardness). However, as far as we know, there are no reports of changes in the physicochemical properties of soft candies when the sugar content is reduced or replaced by agave fructans. In this sense, agave fructan and agave syrup from *A. tequilana* Weber var. azul would be a suitable substitute for sugar and glucose content, obtaining gel formulations with improved physical properties and prebiotic activities compared with the control formulation.

Thus, this study aimed to investigate the changes in physical properties and prebiotic activities of gelatine gels as model systems when agave fructans or agave syrups from *A. tequilana* Weber var. azul were used instead of glucose and sucrose. In order to achieve this purpose, we proposed two objectives: (1) evaluate the physical properties, thermal stability, and texture of gel formulations produced with agave carbohydrates (agave syrups and agave fructans) and (2) evaluate the prebiotic activity of gel formulations in vitro to promote the selective growth of *Lactobacillus* spp., compared with a control gel formulation (sucrose and glucose). This information could help to promote the consumption of agave syrups and agave fructans as functional ingredients in the confectionery industry.

## 2. Results

### 2.1. Physicochemical Properties of Gels

Figure 1a shows the appearance and the colour properties of the gel formulation produced with glucose and fructose (F1) and those formulations made with agave syrups and agave fructans (F2–F4).

Colour impacts the acceptability of food products [20]. Gel formulations were translucent with characteristic colours relative to aspects of their manufacturing processes and ingredients (Figure 1a). Visual colour evaluation is subjective because factors such as size, light, or even the evaluator impact the perception of the colour. For example, F1 gel showed a colourless appearance sample, while agave carbohydrate formulations (F2–F4 gels) presented typical translucent caramel colours of different intensities. The F2 gel showed a medium amber colour, which is typical of agave syrup. The F3 gel displayed a clear light amber colour associated with agave fructan syrup, and the F4 gel had a faint light amber colour correlated with the agave syrup concentration. Table 1 shows the CIEL*a*b*, water activity, moisture content, and pH values of gel formulations.

Figure 1b shows the plot of the CIEL*a*b*values from gel formulations. The high L* values indicated bright samples. Moreover, we can observe that the colour of the formulations was associated with positive values on the +b*-axis and low negative values on the –a*-axis, showing mainly yellow colours. All colour parameters showed significant differences between samples (*p* < 0.05). The results exhibited that the L*values decreased as the +b*value increased. For example, the F2 sample displayed the lowest L*value and the highest +b*value among samples. The agave syrup colours can vary commercially from light to dark amber. The colour of agave syrup is usually linked with mineral salts, phenolic compounds, or processing conditions [6].

On the other hand, impurities in agave fructans are removed by industrial clarification processes, including activated charcoal, membrane filtration, or ultrafiltration [21]. These purification processes of the agave fructans can generate colourless and translucent samples. However, in elevated temperatures, Maillard reactions between gelatine and agave fructans would be responsible for the slightly amber colour in the F3 gel formulation.

Typically, the water content and the water activity of confectionary type-products depend on the cooking temperature and the formulation’s ingredients. Ergun et al. [22] suggested that soft jelly-type products have water activity values between 0.80 and 0.88. The water activity of the foods is related to the microbiological stability, texture, and water migration during storage, where the two last are the most critical parameters in confectionary products [9,22]. In our study, the water content of F2 gel was significantly higher than for the other gel formulations (F2–F4). F3 gel displayed the highest water activity value, associated with a higher capacity of agave fructans to form hydrogen bonds than sugars. Espinosa-Andrews and Rodríguez-Rodríguez [9] suggested that the branched structure of agave fructans retains a high number of water molecules in moderate water activity values inducing a plasticising effect. The water activity and moisture content values obtained in our study were in the range of those reported by Efe et al. [23] for soft gelatine candies using different sweeteners.

Moreover, the pH can impact the mechanical properties of the gels, i.e., low pH can reduce the gel strength of gelatin-based gels, decreasing the hardness of the food product [17]. In our study, gels displayed acidic pH values of around ~3.5 due to incorporating citric acid into gel formulations. Similar pH values were reported by Zhang and Barringer [24], who produced gels using different hydrocolloids such as pectin, starch, and gelatine.

### 2.2. HP-TLC Analysis

Michel-Cuello et al. [25] observed that the hydrolysis rate of agave fructans was influenced by processing parameters, including temperature (>90 °C) and pH (<5.5). In our study, the HP-TLC technique was used to know if the heat process applied during the gel fabrication induced thermal hydrolysis of the agave fructans. Figure 2 displays the HP-TLC silica gel plate and the densitogram for carbohydrate standards and gel formulations (F1, F2, F3, and F4). Fructose and glucose standards were easily identified by the colour of the spot (Figure 2a). Fructose displayed a brown spot, while glucose showed a black spot [26]. Retention factor (Rf) was used to compare samples and identify carbohydrates, and each displayed a particular Rf equal to the distance traveled by the compound divided by the solvent front. Sucrose (S), fructose (F), agave syrup (AS), and agave fructan (AF) were used as standards. For example, a similar Rf was observed between fructose and agave syrup, but agave syrup shares a similar Rf with sucrose (Figure 2b).

Sucrose (S), fructose (F), and agave syrup (AS) showed similar retention factors (Rf). In contrast, the agave fructan (AF) displayed two different spots; a light brown spot correlated with a low fructose concentration and an intense black spot associated with a high polysaccharide content. As expected, gelatine did not display a signal on the HP-TLC silica gel plate (Figure 2b). 

Each gel formulation exhibited a specific carbohydrate profile, i.e., the F1 gel showed a big spot correlated with sucrose and glucose concentrations; F2 displayed an intense brown spot linked with agave syrup, while F3 showed a similar profile agave fructan (AG). Finally, the F4 sample showed spots associated with the fructose and the agave fructan contents. Generally, when the spot of agave fructan migrates to higher Rf values and changes its colour from black to brown, the agavins are hydrolysed. The results obtained in our study show that the thermal process applied during the soft gel fabrication did not induce the thermal hydrolysis of the agave fructans (F3 sample).

### 2.3. Texture Profile Analysis

The texture is the response of the tactile sense to physical stimuli, while texture in the mouth results from contact between the teeth and the food [27]. Texture profile analysis (TPA) is a test to measure food’s textural and mechanical properties, in which controlled compression is applied, and its response is recorded over time [28]. The compression test simulates the chewing process by generating two curves representing different textural food parameters. The texture of soft candies is influenced by the production process, water content/water activity, gelling agent, sugar content, and presence of other components [17]. Figure 3 shows the TPA results of gel formulations (F1, F2, F3, and F4). Hardness, the maximum force required to compress a food, is the main factor impacting the texture of gels, affecting consumers’ acceptance and taste perception [28,29]. The hardness did not show significant differences between the F2, F3, and F4 samples, but the F1 gel did (*p* < 0.05).

Gelatine produces elastic and rubbery gel textures depending on its bloom number, protein, and solute concentration [30]. Previous reports indicated that adding low-molecular saccharides or polyols changes the thermal stability and texture of gel formulations [31,32]. If the gel formulations have high hardness, they require more energy to break during oral processing, negatively impacting consumer acceptance [29]. Gunes et al. [17] reported that the hardness values of soft candies must be between 4.0 and 15.5 N, which are within the hardness values obtained in our study (Figure 3a). The sucrose and glucose substitution with agave fructan and agave syrup increased the hardness of gel formulations, i.e., the F1 gel displayed a softer texture than the other gel formulations (F2–F4 gels). These results suggested that agave ingredients and gelatine struggle with the water molecules for their hydrations, enhancing the elastic properties of gels [30].

Soft candies are semi-solid foods typically characterised by the gumminess texture parameter, calculated by the hardness and cohesiveness. The mean values of gumminess for gel formulations followed the next behaviour (Figure 3b): F2 ^b^ > F3 ^b,c^ > F4 ^c^ > F1 ^a^. The F2 and F3 gel formulations showed significantly higher gumminess values (*p* < 0.05) than the F4 and F1 gels. The F1 gel formulation displayed a weaker gel structure because the mixture of sucrose and glucose decreased the junction zones of the gelatine network [30]. F1 gel showed lower springiness values (Figure 3c) than the agave gels (*p* < 0.05), indicating different elasticity properties when the compressive force was removed [16]. Cohesiveness and chewiness did not show significant differences between the samples (*p* < 0.05). These results indicate that agave carbohydrates can produce soft gelatine gels with similar textural properties to traditional soft gels formulations.

### 2.4. Probiotic Growth Kinetics

Fructans are essential ingredients for functional food industries because of their beneficial effect on human health [6]. Fructans positively impact the host by stimulating the growth of bacteria (probiotics) or limiting the undesirable growth of bacteria in the colon. Probiotics increase the production of metabolites such as short-chain fatty acids, which are essential for gastrointestinal health [33]. *Lactobacillus plantarum*, *Lactobacillus rhamnosus*, and *Lactobacillus paracasei* are microorganisms that form part of human intestine microbiota and typically are used to evaluate in vitro prebiotic effect. Figure 4 shows the growth rates of selected probiotics: *L. plantarum* (4a), *L. paracasei* (4b), and *L. rhamnosus* (4c).

The MRS and MRSm broths were used as positive and negative controls for the growth of *Lactobacillus* strains. Probiotics have complex nutritional requirements, including carbohydrates, peptides, fatty acid esters, amino acids, salts, and vitamins [34]. The growth of probiotics was significantly higher with the gel formulations (F1–F4) than with the MRSm medium (*p* < 0.05). The different growth kinetics showed that each probiotic could consume gel formulations differently. For example, the F1 formulation achieved the highest optical density (OD), followed by F2, F4, and F3 gel formulations for the *L. plantarum* strains (Figure 4a), while *L. paracasei* achieved the highest OD when the F3 formulation was used as a carbon source, followed by F1, F4, and F2 gel formulations (Figure 4b). On the other hand, the F2 gel achieved the maximum OD for *L. rhamnosus*, followed by F1, F4, and F3 samples (Figure 4c). *L. plantarum* strains prefer glucose and sucrose as carbon sources. *Lactobacillus* spp. can ferment hexoses and pentoses, resulting in a faster growth rate because they possess several pathways to metabolise sucrose [35]. According to Table 2, the F1 formulation delayed growth from *L. paracasei* and *L. rhamnosus* strains.

In general, *Lactobacillus* bacteria preferentially consume low-molecular saccharides (glucose and fructose) rather than complex sugars (disaccharides, oligosaccharides, or polysaccharides) [36]. The growth kinetics of probiotics for the F1 sample was similar for sucrose as a carbon source (data not shown). Both *L. plantarum* and *L. rhamnosus* strains prefer to consume the F2 sample as a carbon source over F3 and F4 samples. The F2 sample, composed mainly of fructose from agave syrup, promoted a faster growth of probiotics than the other gel formulations, as evidenced by the μ_max_ values (Table 2). Although the F3 sample was less preferred by *L. plantarum* and *L. rhamnosus*, it was preferred by *L. paracasei*. A similar behaviour was reported previouly for *L. paracasei* because they can metabolise agave fructans [10,37].

Finally, the F4 formulation had similar behaviour for the three *Lactobacillus* tested, showing OD values between 0.79 and 0.93 and µmax between 0.137 and 0.207 h^−1^. Wang et al. [38] reported that the molecular weight of rapeseed polysaccharides as prebiotics significantly affected the proliferation of bifidobacterial strains (*B. adolescentis*, *B. infantis* and *B. bifidum*) and lactobacilli strains (*L. acidophilus*). These results agree with those presented by Huang et al. [39], who described that an appropriate amount of rapeseed carbohydrates promote the growth of beneficial bacteria. Our results showed that the *L. plantarum* preferred the F1 and F2 gel formulations. Mueller et al. [37] reported that the polymerisation degree of fructans had a significant impact on the growth kinetic of probiotics and, thus, on the beneficial effects. The authors reported that fructans with a low polymerisation degree lead to *Lactobacillus’* earlier growth than those with a high degree of polymerisation. These results show that the prebiotic effect of the agave carbohydrates can positively impact the *Lactobacillus* growth in the proximal colon and the distal colon.

## 3. Materials and Methods

### 3.1. Materials

Organic agave fructans 75 °Brix (°B) (Olifructine^TM^ Nutriagaves group, Guadalajara, Jal, Mexico), organic agave syrup 80 °B (dulsweet^TM^ Nutriagaves group, Guadalajara, Jal, Mexico), gelatine 300 Bloom (Progel^TM^, Progel Mexicana SA De CV, León, Gto, Mexico), sucrose, glucose syrup (45 °B), citric acid, sodium citrate, and sodium chloride were used as ingredients in the gels.

### 3.2. Sample Preparation

Four soft gel formulations were prepared. Briefly, 80 g of gelatine was dispersed in water at 20 °C (ratio of 1:2.5) and stored for 30 min. Then, 277 g of glucose syrup was mixed with 416 g of sucrose, and it was heated for 5 min at 105 °C, reaching a soluble solids content of approximately 80 °B. Then, the syrup was removed from the heat, and the gelatine dispersion was added and mixed. Finally, 25.5 g of a solution of citric acid (47% *w/v*), sodium chloride (1.9% *w/v*), and sodium citrate (3.9% *w/v*) solution were added and mixed with the gelatine syrup for 1 min at 80 °C. The final mixture was poured into silicone moulds and kept for 24 h at 20 °C. Silicone moulds were previously coated with mineral oil to prevent the samples from sticking together. The functional gel formulations were produced by total replacing sucrose and glucose content: F2 (675.8 g, agave syrup), F3 (720.9 g, agave fructan syrup), and F4 (338 g, agave syrup and 360.4 g, agave fructan syrup). Gel formulation containing sucrose and glucose was used as control (F1).

### 3.3. Physical Properties

#### Colour, Water Content, Water Activity, and pH of Gels

The colour attributes were measured employing a spectrophotometer (CM5, Konica Minolta Inc., Tokyo, Japan). Samples were carefully loaded in a 10 mm cell. The standard illuminate D65 and standard observer of 10° were used to measure the samples. The results were expressed as CIELAB values L*, a*, and b*. The L*-axis represents the lightness of a colour, with a value of zero (L*= 0) representing black and a value of one hundred (L*= 100) representing white colour. The a*-axis represents the red to green colours, the red in the positive extreme (+a*) and green in the negative extreme (–a*). The b*-axis represents the yellow to blue colours, with yellow in the positive extreme (+b*) and blue in the negative extreme (–b*) [20]. The water content (Wc) of the samples was determined by the AOAC method 950.46 using an oven at 110 °C up to constant weight. The water activity (a_w_) was measured using a dew-point hygrometer at 20 °C (Aqualab 4TE, Decagon Devices Inc, Pullman, WA, USA). The pH of samples was measured using a pH-meter Oakton PH 700 (Oakton Instruments, Vernon Hills, IL, USA). Approximately 1 g of sample was dissolved in 10 mL of distilled water at 50 °C before the pH measurement [16]. For each formulation, three measurements were performed, and the average ± standard deviation was reported.

### 3.4. HP-TLC Analysis

High-performance thin-layer chromatography (HP-TLC) was used to separate and identify carbohydrates by molecular weight. The HP-TLC was carried out using a methodology proposed by Alvarado et al. [40]. Five microliters of samples (20 mg/mL) were put in a pre-coated HP-TLC plate using a sample applicator (CAMAG Linomat, 5 Muttenz, Switzerland) with a 100 L syringe (Hamilton, Bonaduz, Switzerland). An HP-TLC plate nano-sil NH2/UV254 amino-modified was used as the stationary phase (Macherey-Nagel GmbH and Co. KG, Duren, Germany). The mobile phase consisted of n-butanol:methanol:water:acetic acid mixture (50/25/20/1; *v/v/v/v*). The plate was atomised with an ethanol/H_2_SO_4_/anisaldehyde solution (18/1/1; *v*/*v*/*v*) and revealed at 190 °C for 20 min. After chromatography, densitometric scanning was performed using a TLC Scanner 3 (CAMAG Linomat) at 500 nm. Wincats Software v1.4.4.6337 (CAMAG Linomat) operated the scanner. The radiation source was a D2&W lamp, and the scanning speed was 20 mm/s.

### 3.5. Texture Analysis

The texture profile analysis (TPA) was performed using a texturometer TA-XT PLUS (Stable Micro System Co., Ltd., Surrey, England) through two consecutive compression cycles of the samples using a cylindrical probe of 50 mm diameter. The test conditions were: test speed of 0.5 mm s^−1^, trigger point of 0.05 N, a strain of 50%, strain time of 5 s, and charge cell of 5 kg [18]. The textural parameters obtained were hardness, gumminess, chewiness, cohesiveness, and springiness. Ten measurements were performed, and the average ± standard deviation was reported.

### 3.6. Prebiotic Effect

The prebiotic effect was estimated by measuring the growth rate of three probiotic bacteria selected from the collection of the CIATEJ A.C: *Lactobacillus plantarum* LP-115, *Lactobacillus rhamnosus* NH001, and *Lactobacillus paracasei* LPC-37. Briefly, the MRS medium (de Man, Rogosa, and Sharpe medium) was modified by replacing the glucose as a carbon source with the gel formulations (F1, F2, F3, or F4). The probiotic inoculum (1%) was added to the MRS-modified broth mediums and incubated at 37 °C. MRS broth without glucose (MRSm) was used as a negative control [13,14]. The growth rate of bacteria was measured using the Optical density (OD) at 600 nm using Multitask GO UV/VIS spectrophotometer (Thermo Scientific, Waltham, MA, USA). The maximal growth rate (µmax) was estimated by the logistic model using the Software OriginPro2016 (OringinLab Corporation, Northampton, MA, USA) [14]:(1)$$y=\frac{a}{1+be^{-kx}}$$
where *y* is the OD in Arbitrary Units (AU), *a* is the maximal population growth value (AU), *k* is the intrinsic growth rate (h^−1^), *b* is a constant that depends on the initial condition, and *x* is the time (h). Moreover, the maximal growth point was obtained (*t_max_* and *OD_max_*). Three measurements were performed, and the average ± standard deviation was reported.

### 3.7. Statistical Analysis

The data were expressed as means of triplicate samples ± standard deviations. A one-way analysis of variance was applied, and, whenever appropriate, Tukey’s test was used to determine differences among the means (*p* < 0.05). Statistical analyses were performed using Statgraphics Centurion XVI program version 16.1.17 (StatPoint Technologies, Inc., Warrenton, WV, USA).

## 4. Conclusions

Agave fructan and agave syrup are functional ingredients that can be used as healthier substitutes for glucose and sucrose to produce gel formulations, such as soft gels. The HP-TLC demonstrated that agave fructan preserves its chemical structure during the thermal process of soft gels production. These healthier ingredients improve the textural properties and appearance of gel formulations. Our results showed that agave fructans and agave syrups are good carbon sources for selected *Lactobacillus* strains. Agave fructans and agave syrups from *Agave tequilana* Weber var. Azul can be used to promote gastrointestinal health by manipulating human colonic microbiota. The results obtained in this study point to the fact that soft gels containing agave carbohydrates (agave syrups or/and agave fructans) can be used to produce functional confectionery products, but according to the prebiotic effect and textural properties, the F2 and F4 gel formulations displayed the best techno-functional properties. However, it is necessary to develop new research to know the consumers’ sensory acceptance of gel formulation and other health benefits such as glycaemic index modulation.

## Figures and Tables

**Figure 1 molecules-27-04902-f001:**
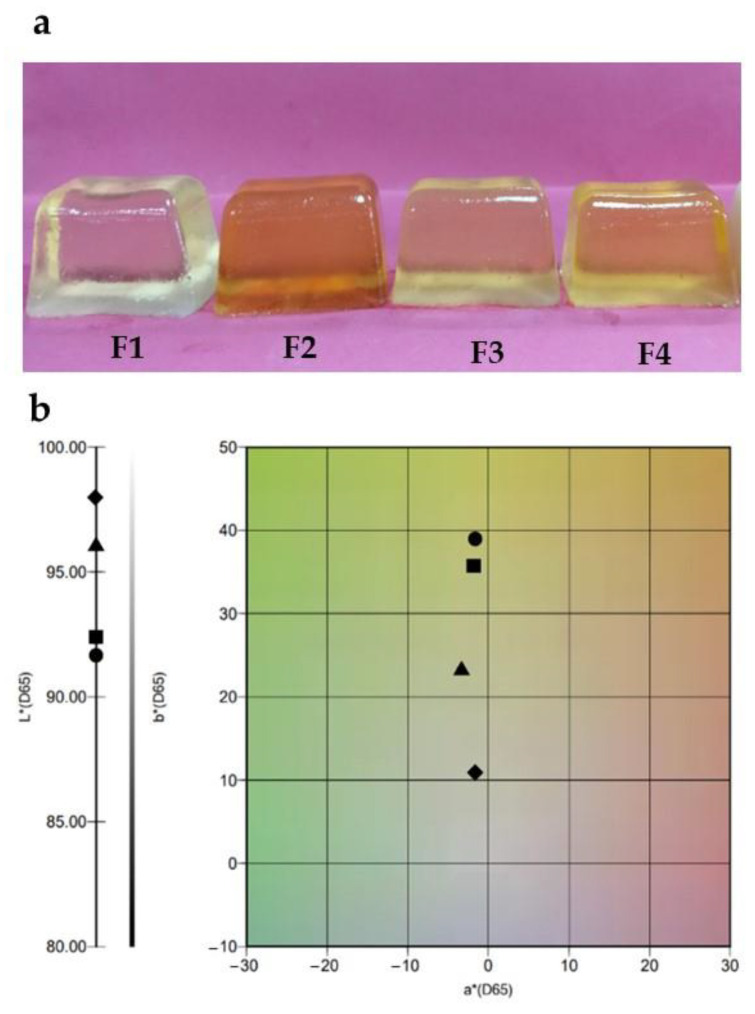
(**a**) The appearance of the gel formulations: F1 (control formulation), F2 (100% agave syrup), F3 (100% agave fructan), and F4 (50% agave syrup and 50% agave fructan); (**b**) CIEL*a*b*spectrum of gels formulation: F1 (rumble), F2 (circle), F3 (triangle), and F4 (square).

**Figure 2 molecules-27-04902-f002:**
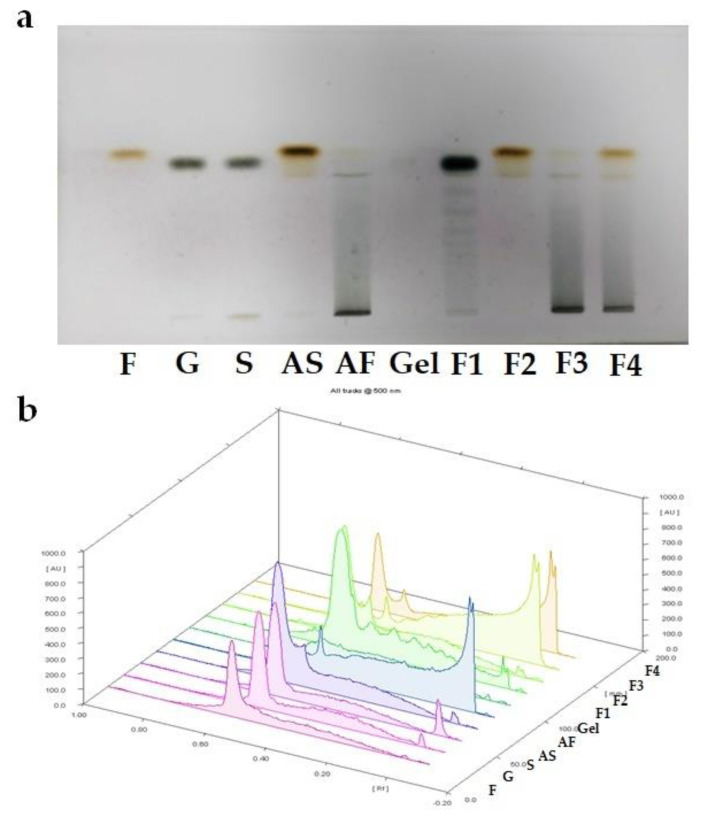
(**a**) HP-TLC silica gel plates of gel formulations; (**b**) densitogram obtained from carbohydrates standards and gel formulations: Fructose (F), glucose (G), sucrose (S), agave syrup (AS), agave fructan (AF), gelatine (Gel), and gel formulations: F1 (control formulation), F2 (100% agave syrup), F3 (100% agave fructan), and F4 (50% agave syrup and 50% agave fructan).

**Figure 3 molecules-27-04902-f003:**
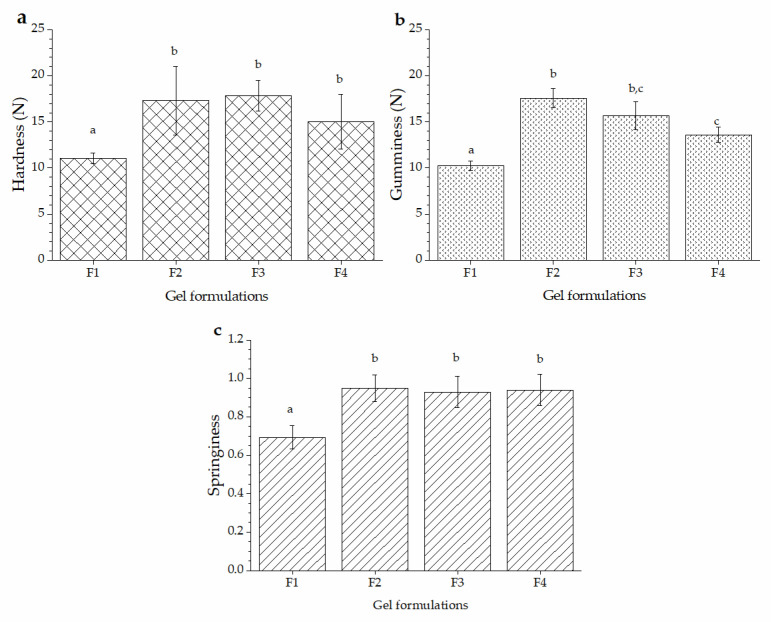
TPA analysis of gel formulations: (**a**) Hardness, (**b**) gumminess, and (**c**) springiness of the F1 (control formulation), F2 (100% agave syrup), F3 (100% agave fructan), and F4 (50% agave syrup and 50% agave fructan). ^a,b,c^ Different letters indicate a significant difference (*p* < 0.05).

**Figure 4 molecules-27-04902-f004:**
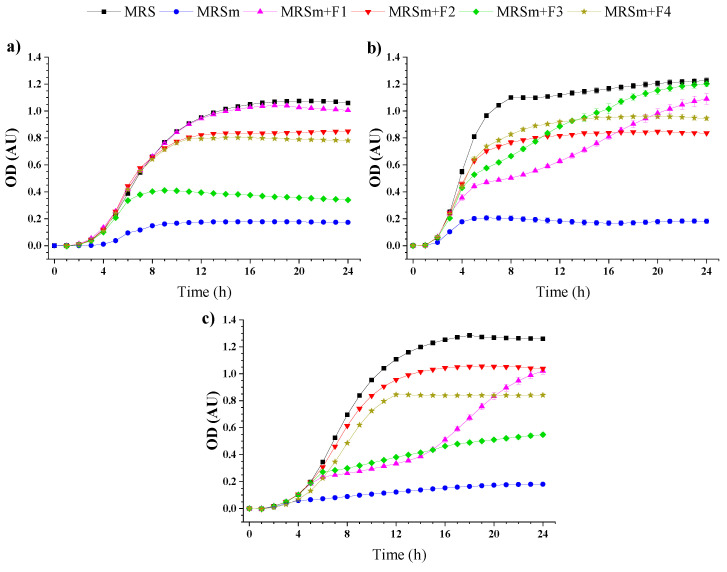
Growth kinetics of (**a**) *Lactobacillus plantarum*, (**b**) *Lactobacillus paracasei*, and (**c**) *Lactobacillus rhamnosus* in MRS medium with glucose (square, MRS), without glucose (circle, MRSm), and replacing glucose as a carbon source with the gel formulations: MRSm + F1 (triangle, control formulation), MRSm + F2 (inverted triangle, 100% agave syrup), MRSm + F3 (rumble, 100% agave fructan), and MRSm + F4 (star, 50% agave syrup and 50% agave fructan).

**Table 1 molecules-27-04902-t001:** Physical properties of the gel formulations.

	Gel Formulations			
Parameter	F1	F2	F3	F4
L*	97.98 ± 0.05 ^a^	92.27 ± 1.18 ^b^	96.06 ± 0.06 ^c^	91.73 ± 0.51 ^d^
a*	−1.73 ± 0.04 ^a^	−1.90 ± 0.12 ^b^	−3.41 ± 0.07 ^c^	−1.58 ± 0.23 ^d^
b*	10.86 ± 0.32 ^b^	35.86 ± 0.13 ^b^	23.08 ± 0.10 ^c^	39.09 ± 0.52 ^d^
Water content (g/100 g)	38.98 ± 0.14 ^a^	41.86 ± 0.28 ^b^	40.37 ± 0.38 ^c^	40.25 ± 0.06 ^c^
Water activity	0.890 ± 0.005 ^a,b^	0.823 ± 0.001 ^c^	0.929 ± 0.001 ^a^	0.855 ± 0.019 ^b,c^
pH	3.50 ± 0.01 ^a,b^	3.51 ± 0.01 ^a^	3.43 ± 0.01 ^c^	3.45 ± 0.01 ^b,c^

^a,b,c,d^ Different letters in the same row indicate a significant difference (*p* < 0.05). F1 (control formulation), F2 (100% agave syrup), F3 (100% agave fructan), and F4 (50% agave syrup and 50% agave fructan). The L* parameter represents the lightness of a colour, the a* parameter represents the red (positive values) to green (negative values) colours, and the b* parameter represents the yellow (positive values) to blue colours (negative values).

**Table 2 molecules-27-04902-t002:** Maximum growth rates for *Lactobacillus* strains.

	Maximum Growth Rates, µ_max_ (h^−1^)		
Sample	*L. plantarum*	*L. paracasei*	*L. rhamnosus*
MRS	0.155 ± 0.001 ^ab^	0.280 ± 0.013 ^a^	0.176 ± 0.001 ^a^
MRSm	0.042 ± 0.001 ^e^	0.076 ± 0.006 ^e^	0.024 ± 0.000 ^f^
F1	0.152 ± 0.002 ^b^	0.049 ± 0.005 ^f^ 0.146 ± 0.009 ^d,^*	0.068 ± 0.001 ^e^ 0.084 ± 0.003 ^d,^*
F2	0.162 ± 0.002 ^c^	0.199 ± 0.007 ^b,c^	0.152 ± 0.002 ^b^
F3	0.117 ± 0.001 ^d^	0.054 ± 0.002 ^f^ 0.184 ± 0.002 ^c,^*	0.024 ± 0.001 ^f^ 0.085 ± 0.002 ^d,^*
F4	0.157 ± 0.002 ^a^	0.205 ± 0.005 ^b^	0.137 ± 0.002 ^c^

^a,b,c,d,e,f^ Different letters in the same row indicate a significant difference (*p* < 0.05). * Second maximum growth rate. F1 (control formulation), F2 (100% agave syrup), F3 (100% agave fructan), and F4 (50% agave syrup and 50% agave fructan).

## Data Availability

The data presented in this study are available on request from the corresponding author.

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
