# Peer review of "Physical Properties and Prebiotic Activities (*Lactobacillus* spp.) of Gelatine-Based Gels Formulated with Agave Fructans and Agave Syrups as Sucrose and Glucose Substitutes"

_molecules, 2022, doi:10.3390/molecules27154902_

Round 1
Reviewer 1 Report
The manuscript submitted by Rogelio Rodríguez-Rodríguez et al. deals with the characterization of physical properties and prebiotic activities of gelatine-based gels formulated with agave fructans and agave syrups as substitutes for sucrose and glucose. The topic is quite original and fully falls within the scope of the special issue. However the study is based on a very simple experimental plan; indeed, it would have been very interesting to have additional information on the prebiotic effect, which was instead analyzed only through the evaluation of the growth rate of selected strains.
Specific comments:
- The abstract must be integrated by reporting the quantities of agave fructans and agave syrups used in the preparation of the formulations. Furthermore it should be state if there was a preference for agave fructans or agave syrups.
- L86-94: The hypothesis from which the experimentation originates is not clear.
- In Tables footnotes and Figure captions (when necessary) must be reported the explanation for the notations F1, F2, F3 and F4.
- L274-275: “A citric acid, sodium chloride, and sodium citrate solution was added…”. In what quantity was the solution added?
- L279-280: F1? The presence of a “Control” condition must be clearly stated.
- L270-280: How many soft gels were prepared for each formulation? How many samples were then used for the analysis?
- L310-313: What parameters were evaluated with this analysis?
- L316-317: Indicate the reason why these three strains were selected.
- L334-335: “All measurements were reported as means ± standard deviations of three independent experiments”. Also here it must be clearly indicated how many samples were analyzed in each experiment.
- L340-348: In the conclusions it should be indicated whether there is a preference in the use of Agave fructans or agave syrups, or it must be clearly stated that the use of these ingredients confers overlapping properties to the final products.
Author Response
Response to Reviewer 1 Comments
The authors would like to thank the reviewers for providing critical comments and suggestions to improve the quality of the manuscript. We have revised the manuscript accordingly and included a detailed list of responses below. For your ease of reading, we have modified the manuscript in red.
- The manuscript submitted by Rogelio Rodríguez-Rodríguez et al. deals with the characterization of physical properties and prebiotic activities of gelatine-based gels formulated with agave fructans and agave syrups as substitutes for sucrose and glucose. The topic is quite original and fully falls within the scope of the special issue. However the study is based on a very simple experimental plan; indeed, it would have been very interesting to have additional information on the prebiotic effect, which was instead analyzed only through the evaluation of the growth rate of selected strains.
Reply: Thank you for your comments. Our research group focuses on the impact of bioactive molecules and foods on gut microbiota. Our first analysis consists of evaluating the growth rate of selected probiotic strains against an ingredient to know/evaluate their affinity. These results help us make decisions for the design of expensive and time-consuming in vivo or ex vivo studies, which are expected to carry out in the next year.
- The abstract must be integrated by reporting the quantities of agave fructans and agave syrups used in the preparation of the formulations. Furthermore it should be state if there was a preference for agave fructans or agave syrups.
Reply: Thank you for your comments. The percentage of the agave fructans and agave syrups used in the preparation of the formulations was added in the abstract: The sugars (sucrose and glucose) were substituted by agave carbohydrates (agave fructans and agave syrups), obtaining the subsequent gel formulations: 100% agave syrups (F2), 100% agave fructans (F3), and 50% agave syrups-50% agave fructans (F4). Unsubstituted gel formulación was used as a control (F1).
The mass of agave fructans and agave syrups used in the preparation of the formulations was described in detail in the sample preparation section: Four soft gel formulations were prepared. Briefly, 80 g of gelatine was dispersed in water at 20 °C (ratio of 1:2.5) and stored for 30 min. Then, 277 g of glucose syrup was mixed with 416 g of sucrose, and it was heated for 5 min at 105 °C, reaching a soluble solids content of approximately 80 °B. Then, the syrup was removed from the heat, and the gelatine dispersion was added and mixed. 25.5 g of a solution of citric acid (47 % w/v), sodium chloride (1.9 % w/v), and sodium citrate (3.9 % w/v) solution were added and mixed with the gelatine syrup for 1 min at 80 °C. The final mixture was poured into silicone moulds and kept for 24 h at 20 °C. Silicone moulds were previously coated with mineral oil to prevent the samples from sticking together. The functional gel formulations were produced by total replacing sucrose and glucose content: F2 (675.8 g, agave syrup), F3 (720.9 g, agave fructan syrup), and F4 (338 g, agave syrup, and 360.4 g, agave fructan syrup). Gel formulation containing sucrose and glucose was used as control (F1).
In our opinion, to respond to the question of which gel formulation is preferred? We require developing a consumer preferences study. This study was not included because it needed many participants (n=300) to generate significant statistical data. However, our preliminary analysis, made with a small number of semi-trained judges (n=8), indicated that the gel produced with agave syrup (F2) had a less acidic flavor than the control (F1) but high than the agave fructans formulation (F3). Also, F2 was the brittlest, sweetest, and chewiest gel.
- L86-94: The hypothesis from which the experimentation originates is not clear.
Reply: We apologize for the misunderstanding. The hypothesis from which the experimentation originates was clearly indicated in the introduction section.
Agave fructan and agave syrup from A. tequilana Weber var. azul will be a suitable substitute for sugar and glucose content, obtaining gel formulations with improved physical properties and prebiotic activities compared with control formulation.
- In Tables footnotes and Figure captions (when necessary) must be reported the explanation for the notations F1, F2, F3 and F4.
Reply: Thank you for your comments. The explanation for the notations F1, F2, F3 and F4 were added.
F1 (control formulation), F2 (100% agave syrup), F3 (100% agave fructan), and F4 (50% agave syrup and 50% agave fructan).
- L274-275: “A citric acid, sodium chloride, and sodium citrate solution was added…”. In what quantity was the solution added?
Reply: Thank you for your comments. The concentration of the citric acid, sodium chloride, and sodium citrate were added in the sample preparation section.
25.5 g of a solution of citric acid (47 % w/v), sodium chloride (1.9 % w/v), and sodium citrate (3.9 % w/v) solution were added and mixed with the gelatine syrup for 1 min at 80 °C.
- L279-280: F1? The presence of a “Control” condition must be clearly stated.
Reply: Thank you for your comments. The control formulation was stated in the samples preparation section and when it was necessary throughout the manuscript.
Gel formulation containing sucrose and glucose was used as control (F1).
- L270-280: How many soft gels were prepared for each formulation? How many samples were then used for the analysis?
Reply: Thank you for your comments. Forty soft gels of each formulation (F1, F2, F3, and F4) were prepared. The number of samples used in each analysis was: ten samples for texture analysis, while three samples were used for physical properties, prebiotic, and HP-TLC analysis.
- L310-313: What parameters were evaluated with this analysis?
Reply: Thank you for your comments. The textural parameters obtained were hardness, gumminess, chewiness, cohesiveness, and springiness. Hardness, gumminess, and springiness showed a significant difference between samples.
- L316-317: Indicate the reason why these three strains were selected.
Reply: Thank you for your comments. We selected Lactobacillus plantarum, Lactobacillus rhamnosus, and Lactobacillus paracasei because these strains are part of human intestine microbiota and typically are used to evaluate in vitro prebiotic effect. The manuscript was modified as a follow: Human gut microbiota comprises diverse microorganisms populations; mainly, the small intestine is colonized primarily by Lactobacillaceae and Enterobacteriaceae [11]. These population strains in the human gut depend on dietary habits, race, stress, medication, host genetics, toxin exposure, and pathogen invasion. There have been reported that the Lactobacillus plantarum, Lactobacillus rhamnosus, and Lactobacillus paracasei are used to prevent and treat diverse intestinal diseases in human health [12, 13]
- L334-335: “All measurements were reported as means ± standard deviations of three independent experiments”. Also here it must be clearly indicated how many samples were analyzed in each experiment.
Reply: We apologize for the misunderstanding. The number of samples analyzed in each experiment was clearly indicated in the methodology section.
- L340-348: In the conclusions it should be indicated whether there is a preference in the use of Agave fructans or agave syrups, or it must be clearly stated that the use of these ingredients confers overlapping properties to the final products.
Reply: Thanks for your suggestion. In this work, we focus on the physical properties and prebiotic activity of agave gel formulations. The samples show excellent physical and prebiotic activities. However, it is necessary developing new investigations about their impact on human health; for example, we are very interested in knowing, how the consumption of these agave gel formulations modifies the glycemic index in human populations? or if, these agave gel formulations can be used to modify the absorption of the nutrients? or if these can favour the probiotic growth in the human population?
Reviewer 2 Report
This study evaluated agave fructans and agave syrup as healthy alternatives to sucrose and glucose by measuring the physicochemical and biological properties of the gels, which may provide help in the development of new healthy foods. However, there are still some problems in the manuscript that need to be revised, and the following are my suggestions.
1. Line 70: The word “represent” is suggested to be revised to “provide”.
2. Lines 143-150: This paragraph cited many references, and I suggest revising the expression of sentences.
3. Lines 160-161: It would be better if the author could explain the "retention factors" and describe what similar retention factors represent.
4. Lines 185-199: As you said, the hardness values of soft candies must be within a certain range, but the hardness of F2, F3 and F4 are all high. Will this have a negative impact on consumer acceptance?
5. Is there any basis for using the same content of agave syrups and agave fructans in F4? Does the author consider exploring the impact of their different proportions in the future work?
6. Line 326: Please cite appropriate reference here.
7. Line 41-44: In addition to this reference, I suggest adding at least one more example.
8. This manuscript cited a very large number of references, and if possible, some of them should be omitted.
Author Response
Response to Reviewer 2 Comments
The authors would like to thank the reviewers for providing critical comments and suggestions to improve the quality of the manuscript. We have revised the manuscript accordingly and included a detailed list of responses below. For your ease of reading, we have modified the manuscript in red.
- This study evaluated agave fructans and agave syrup as healthy alternatives to sucrose and glucose by measuring the physicochemical and biological properties of the gels, which may provide help in the development of new healthy foods. However, there are still some problems in the manuscript that need to be revised, and the following are my suggestions.
Reply: Thank you very much for your comments. The following comments are addressed accordingly.
- Line 70: The word “represent” is suggested to be revised to “provide”.
Reply: We appreciate your suggestion. The word “represent” was changed to “provide”.
- Lines 143-150: This paragraph cited many references, and I suggest revising the expression of sentences.
Reply: We appreciate your suggestion. The paragraph was revised and corrected.
- Lines 160-161: It would be better if the author could explain the "retention factors" and describe what similar retention factors represent.
Reply: Retention factor definition was added and the sentence was rewritten.
Retention factor (Rf) is used to compare samples and identify carbohydrates. Each of them displayed a particular Rf equal to the distance traveled by the compound divided by the solvent front. Sucrose (S), fructose (F), agave syrup (AS), and agave fructans (AF) were used as standards. For example, similar Rf were observed between fructose and agave syrup but also agave syrup share similar Rf with sucrose (Fig. 2b)
- Lines 185-199: As you said, the hardness values of soft candies must be within a certain range, but the hardness of F2, F3 and F4 are all high. Will this have a negative impact on consumer acceptance?
Reply: Thank you for your comments. The results displayed that the hardness did not show significant differences between the F2, F3, and F4 samples, but the F1 gel did If the gel formulations have high hardness, they will require more energy to break during oral processing, negatively impacting consumer acceptance.. Therefore, soft candies too hard can have a negative impact on consumer acceptance. Gunes et al. [17] reported that the hardness values of soft candies must be between 4.0 and 15.5 N, which are within the hardness values obtained in this study (Figure 3a).
- Is there any basis for using the same content of agave syrups and agave fructans in F4? Does the author consider exploring the impact of their different proportions in the future work?
Reply: Thank you for your comments. Our first analysis evaluated the changes in physical properties and prebiotic activities of gelatine gels as model systems when agave fructans or agave syrups were used instead of glucose and sucrose. Interestingly, there are no reports of changes in the physicochemical properties of soft candies when the sugar content is reduced or replaced by agave fructans. Thus, we decided to formulate four soft candies: F1 (control formulation), F2 (100% agave syrup), F3 (100% agave fructan), and F4 (50% agave syrup and 50% agave fructan).
These results help us make decisions for the design of new formulations containing different proportions of prebiotics, which are expected to carry out in the next year.
- Line 326: Please cite appropriate reference here.
Reply: We appreciate your suggestion. We cited correctly the reference Michel-Barba et al. 2019.
- Line 41-44: In addition to this reference, I suggest adding at least one more example.
Reply: We appreciate your suggestion. Two examples were added.
Santiago-García et al. [4] reported that adding agave fructans to fat-reduced yogurts decreased the syneresis and increased water retention capacity and viscosity compared with the yogurt fat-reduced control. González-Herrera et al. [5] produced a snack from dehydrated apple supplemented with agave fructans exerting a probiotic effect by increasing short-chain fatty acid production in mice.
- This manuscript cited a very large number of references, and if possible, some of them should be omitted.
Reply: We appreciate your suggestion. The references in the manuscript were meticulously revised, and the references most significant were cited.
Round 2
Reviewer 1 Report
The authors adequately addressed all criticisms raised during the review process. However, a small question still remains open, probably deriving from a misunderstanding about one of my requests.
Specifically authors were asked to indicate in the abstract and in the conclusions whether there was a preference for agave fructans or agave syrups. Obviously I didn't mean from the consumer's point of view, I meant which formulation had the best results (F2, F3 or F4?). In the current version of the paper it is reported that: "The results showed that agave syrups and agave fructans display great potential as functional confectionery products (abstract)" and "The results obtained in this study point to the fact that soft gels containing agave syrup and agave fructan display great potential as a functional food (Conclusions)". Does this mean that the best formulation is F4 in which both matrices are present? This aspect must be specified.
Author Response
Response to Reviewer 1 Comments (Round 2)
The authors would like to thank the reviewer for providing critical comments and suggestions to improve the quality of the manuscript. We have revised the manuscript accordingly and included the response below. For your ease of reading, we have modified the manuscript in red.
- The authors adequately addressed all criticisms raised during the review process. However, a small question still remains open, probably deriving from a misunderstanding about one of my requests.
Specifically authors were asked to indicate in the abstract and in the conclusions whether there was a preference for agave fructans or agave syrups. Obviously I didn't mean from the consumer's point of view, I meant which formulation had the best results (F2, F3 or F4?). In the current version of the paper it is reported that: "The results showed that agave syrups and agave fructans display great potential as functional confectionery products (abstract)" and "The results obtained in this study point to the fact that soft gels containing agave syrup and agave fructan display great potential as a functional food (Conclusions)". Does this mean that the best formulation is F4 in which both matrices are present? This aspect must be specified.
Reply: We apologize for the misunderstanding. Based on the prebiotic effect, physical, and textural properties, the F2 (100% agave syrup) and F4 (50% agave syrup/50% agave fructan) gel formulations displayed the best techno-functional properties to produce gel soft candies.
Reviewer 2 Report
All the questions have been well addressed.
Author Response
The authors would like to thank the reviewer for providing critical comments and suggestions to improve the quality of the manuscript.